# A Simple In Vitro Test to Select Stools for Fecal Microbiota Transplantation to Limit Intestinal Carriage of Extensively Drug-Resistant Bacteria

**DOI:** 10.3390/microorganisms11112753

**Published:** 2023-11-11

**Authors:** Angélique Salandre, Johanne Delannoy, Marie Thérèse Barba Goudiaby, Frédéric Barbut, Muriel Thomas, Anne-Judith Waligora-Dupriet, Nathalie Kapel

**Affiliations:** 1INSERM UMR-S1139, Faculty of Pharmacy, Université Paris Cité, F-75006 Paris, France; angelique.salandre@edu.mnhn.fr (A.S.); johanne.delannoy@u-paris.fr (J.D.); m.goudiaby@globalbiot.fr (M.T.B.G.); frederic.barbut@aphp.fr (F.B.); anne-judith.waligora@u-paris.fr (A.-J.W.-D.); 2Infection Control Unit, APHP, Saint-Antoine Hospital, F-75012 Paris, France; 3Paris Center for Microbiome Medicine (PaCeMM), Fédération Hospitalo-Universitaire, F-75011 Paris, France; muriel.thomas@inrae.fr; 4Micalis Institute, AgroParisTech, INRAE, Université Paris-Saclay, F-78350 Jouy-en-Josas, France; 5Department of Coprology, APHP, Pitié-Salpêtrière Hospital, F-75013 Paris, France

**Keywords:** fecal microbiota transplant, XDR bacteria, *Klebsiella pneumoniae* OXA-48, *Enterococcus faecium* VanA, screening method, pre-clinical model, human microbiota-associated mice

## Abstract

Treatment options for multidrug-resistant bacterial infections are limited and often ineffective. Fecal microbiota transplantation (FMT) has emerged as a promising therapy for intestinal multidrug-resistant bacterial decolonization. However, clinical results are discrepant. The aim of our pilot study was to evaluate the screening performance of a simple diagnostic tool to select fecal samples that will be effective in decolonizing the intestine. Fecal samples from 10 healthy subjects were selected. We developed an agar spot test to evaluate their antagonistic activity toward the growth of VanA *Enterococcus faecium* and OXA-48-producing *Klebsiella pneumoniae*, two of the most serious and urgent threats of antibiotic resistance. Most fecal samples were able to limit the growth of both bacteria in vitro but with large inter-individual variation. The samples with the highest and lowest antagonistic activity were used for FMT in a mouse model of intestinal colonization. FMT was not successful in reducing intestinal colonization with VanA *Enterococcus faecium*, whereas FMT performed with the fecal sample showing the highest activity on the agar spot test was able to significantly reduce the intestinal colonization of mice with *Klebsiella pneumoniae* OXA-48. The agar spot test could thus serve as a reliable screening tool to select stool samples with the best potential to eradicate/reduce multidrug-resistant bacteria carriage after FMT.

## 1. Introduction

Bacterial infections caused by multidrug and extensively drug-resistant (XDR) pathogens represent a major public health threat that undermines antimicrobial therapy and its fundamental role in modern medicine. Drug-resistant community-onset and nosocomial infections have increased over the last decade and epidemiological data suggest that up to 4.95 million (95% CI: 3.62–6.57) deaths were associated with bacterial antimicrobial resistance in 2019, including 1.27 million (95% CI: 0.911–1.71) deaths directly attributable to bacterial antimicrobial resistance [1].

In this context, one of the most worrying problems is the dramatic increase in the number of infections caused by XDR bacteria, including carbapenem-resistant *Enterobacteriaceae* (CRE) and/or glycopeptide-resistant *Enterococci* (GRE),with few effective therapeutic options remaining in the armamentarium of clinicians. There is thus a striking lack of new antimicrobial agents while the dissemination of these XDR bacteria is accelerating. Even if new agents were discovered, the lead time is considerable as there are no truly new agents expected on the market in the short or medium term. Alternative effective measures to contain resistance and limit the spread of XDR bacteria are therefore urgently needed and represent a public health priority. But, this effort is hindered by the fact that the gut microbiota is the main reservoir of XDR bacteria and that intestinal carriage can persist for months to years, putting carriers at risk of recurrent infections and representing a reservoir for transmission [2,3]. Several decolonization strategies for CRE and GRE carriers have been assessed but have only shown moderate efficacy [4,5].

Fecal microbiota transplantation (FMT) involves the gastrointestinal application of stool from healthy donors with the aim to restore the protective microbiome of natural colonic microbiota. FMT has been shown to be highly effective for the treatment of recurrent *Clostridioides difficile* infections, with an overall cure rate of over 90%. This procedure appears to be safe and well tolerated. FMT is thus now recommended by several scientific societies, including the European Society for Clinical Microbiology and Infectious Diseases (ESCMID), to treat severe CDI episodes or to prevent recurrences and is now becoming widely available in many countries [6,7,8]. Recent studies have also reported results supporting the role of FMT in the primary treatment of severe *C. difficile* infection, with an overall cure rate of over 85% [9]. Interestingly, intestinal dysbiosis, which is responsible for CDI, has also been linked to the persistence of XDR bacteria carriage. Indeed, persistent carriage of XDR bacteria is favored by previous antibiotic consumption and the destruction of the colonization resistance functions of the microbiota. Correcting dysbiosis and restoring the colonization resistance capacities of the microbiota through FMT may therefore be a potential solution to eradicate the XDR bacteria colonizing the gut. This new therapeutic approach has thus been suggested as a decolonization strategy for patients carrying intestinal multidrug-resistant bacteria. However, case reports and non-controlled studies have shown conflicting results, and meta-analyses reported decolonization rates ranging from 20 to 90% [10,11]. The only randomized clinical study showed slightly (but not significantly) lower extended spectrum β-lactamase (EBSL-E)/CRE carriage during the follow-up of patients compared to controls. Although promising, this study was hindered by the failure to achieve the planned sample size, thus precluding any firm conclusions [12].

Another issue concerning clinical trials of FMT is the inter-individual variability of the microbiota, which could account for the reported varying efficacy. This has been clearly shown in studies evaluating the performance of FMT in the maintenance of remission in patients with ulcerative colitis. Indeed, in the study by Moayyedi et al., therapeutic success was essentially achieved only with fecal transplants originating from a single donor [13]. This has led to the notion of “super-donors” [14], raising the question of whether inter-individual diversity of the microbiota could lead to metabolic specificity that has not yet been identified, resulting in an enhanced efficacy of FMT.

We addressed this question by developing a simple agar spot test to confirm and evaluate such inter-individual variability and identify in vitro fecal samples with promising efficacy against the growth of VanA *Enterococcus faecium* (VRE) and OXA-48-producing *Klebsiella pneumoniae* (OXA-48 CRE), two of the most common XDR bacterial species that cause infections in hospitalized patients [15]. We then used this test in a pilot study to select samples with high or low antagonist activity against these two XDR bacteria and to evaluate the efficacy of FMT performed on three consecutive days with the selected fecal samples to limit intestinal colonization with these bacteria in mice.

## 2. Methods

We developed an agar spot test to measure the antagonistic effects of fecal samples on the growth of VRE and OXA-48 CRE in vitro. In this pilot study, we used this agar spot test to measure the antagonistic effects of 10 samples and, using a mouse model of intestinal colonization, evaluated the ability of 4 samples selected for their high or low antagonistic effect against VRE and OXA-48 CRE to limit their carriage after FMT.

### 2.1. Preparation of Stool Samples

Fecal samples were collected from the routine workload of the Department of Coprology (Pitié-Salpêtrière Hospital, 75013 Paris, France) for the follow-up of patients. The use of leftover stools is approved by the French public health organization. We selected samples from patients with no gastrointestinal disorders, no acute or chronic diseases that could alter the intestinal microbiota, and no history of antibiotic treatment or traveling in the previous 3 months. Samples obtained within 2 h of collection were used for this study. Fecal samples were immediately homogenized at the time of reception, then diluted 1:5 at room temperature in aerobic conditions in 10% glycerol, used as a cryoprotectant, and stored at −80 °C as recommended by the International Consensus Conference on Stool Banking for Fecal Microbiota Transplantation in Clinical Practice [16] and used in clinical studies [12,17]. We have previously shown that this procedure preserves bacterial viability and fermentative activities for both aerobic and anaerobic bacteria, even extremely oxygen-sensitive bacteria [18].

We confirmed the absence of mucosal inflammation by measuring fecal calprotectin, and only samples with fecal concentrations below 50 µg/g were used for this study.

Finally, 10 samples were selected for this study.

### 2.2. Bacterial Strains

Both the carbapenemase (OXA-48)-producing *K. pneumoniae* and vancomycin-resistant (VanA) *Enterococcus faecium* clinical isolates were kindly provided by Prof. F. Barbut (collection of the Infection Control Unit, Saint-Antoine Hospital, Paris 75012, France). 

### 2.3. In Vitro Evaluation of the Anti-XDR Bacteria Growth Activity of Human Fecal Microbiota

The agar spot test was adapted from our previous study on *Clostridioides difficile* (18). The capacity of the fecal microbiota suspensions to inhibit the growth of OXA-48 CRE or VRE was evaluated by measuring the diameter of inhibition of the culture relative to that of positive and negative biological controls (*Lactobacillus* spp. S625 and *E. coli* E213, respectively) and reference antibiotics (colistin and tigecycline for OXA-48 CRE and VRE, respectively). An overnight culture of each bacterial suspension (20 μL) was used as a control. An equal volume of each fecal suspension of thawed samples was deposited on cellulose disks and placed on Wilkins–Chalgren agar plus blood for overnight incubation at 37 °C in an anaerobic atmosphere. At the same time, the XDR bacteria, OXA-48 CRE and VRE, were cultured in trypticase soja broth for 24 h at 37 °C under aerobic conditions. The next day, agar plates were covered with Muller–Hinton agar (15 mL) containing either 1 mL of an OXA-48 CRE (McFarland = 0.5) or VRE (McFarland = 1) culture. The reference antibiotic disk (colistin 50 µg or tigecycline 15 µg) was then added. Plates were incubated for another overnight period under aerobic conditions at 37 °C, and the zones of inhibition of the XDR bacteria culture around each spot were measured. Results (from 3 independent experiments) are expressed as the ratio between the inhibition diameters of pure bacteria or stool samples and the reference antibiotic control. Fecal samples with a ratio between the inhibition diameters of the stool samples and the reference antibiotic control ≥ 1 were considered to have a positive antagonistic effect.

### 2.4. Evaluation of the Inhibitory Effect of FMT in an XDR Intestinal Carriage Mouse Model

Animal experiments were performed in accordance with the European guidelines for the care and use of laboratory animals.

The model of intestinal carriage of OXA-48 CRE or VRE was adapted from the procedure of Ubeda et al. and Caballero et al. [19,20]. Animal experiments were approved by the Regional Council of Ethics for animal experimentation C2EA-45 with authorization APAFIS: #31901-2021031614025559 v3.

Six- to eight-week-old female C57BL/6 mice (Charles River, France) were housed at the PharmAnima platform (authorization E75-06-02) in groups of four per cage and maintained under a 12 h light–dark cycle. The food (standard chow), water, bedding, and cages were autoclaved. Cage changes and assessment of the physical condition and behavior of the animals were performed under a laminar flow hood.

After arrival at the animal facility, mice were housed for one week to allow for acclimatization. The health status was checked to verify the absence of XDR bacteria in the fecal microbiota. Animals were then treated for 8 days with amoxicillin (0.5 g/L) (Sigma Aldrich, France) dissolved in the drinking water (ad libitum), with a change every 2 days to avoid any alteration of the antibiotic.

One day before the end of the antibiotic treatment, mice were orally infected with 10^6^ colony-forming units (CFU) of either the OXA-48 CRE or VRE strain by gavage in a total volume of 0.2 mL of phosphate-buffered saline (PBS). The gavage solutions used for OXA-48 CRE and VRE were then stored at −80 °C. One day after the end of the antibiotic treatment, i.e., 48 h after oral gavage with OXA-48 CRE or VRE (D2), mice underwent FMT by gavage with 12.5 mg native stool diluted in 200 µL of PBS (n = 8 per stool sample; two stools selected for each model of XDR bacteria carriage, one with a high degree of inhibition toward the growth and one with a low level of inhibition, as measured by the agar spot screening test performed in vitro). FMT was performed on 3 consecutive days (i.e., D2, D3, and D4 post-gavage). Mice from the control group received an oral gavage with PBS (200 µL, n = 8). 

All animals were then observed twice a week until euthanasia for symptoms and mortality, their weight was recorded, and feces were collected. At D14 or D15 post-gavage with either OXA-48 CRE or VRE, mice were euthanized by intraperitoneal injection of a mixture of ketamine (10 mg/mL) and xylazine (1.2 mg/mL) followed by cervical dislocation, and the colon contents were collected for evaluation of the bacterial load.

### 2.5. Monitoring of XDR Intestinal Carriage

To monitor the intestinal OXA-48 CRE or VRE load, fecal pellets were collected daily from D2 (24 h post-gavage) to D4 (last FMT) and then twice a week until euthanasia of the mice (D14 or D15), i.e., 8 spot samplings. After collection, pellets were immediately diluted 1:10 in a Brain Heat Infusion medium containing 10% glycerol and stored at −80 °C. After thawing, the gavage solution and fecal samples were homogenized and then serially diluted (from 10^−1^ to 10^−6^) in sterile peptone water before spreading onto selective chromogenic agar (ChromID VRE and ChromID OXA- 48, Biomérieux, France). Plates were incubated at 37 °C in an aerobic atmosphere for 24 h. Bacterial counts are expressed as the log CFU/g of feces. The limit of detection (LOD) was 3 log_10_ CFU/g of feces.

### 2.6. Statistical Analysis

All results are presented as the mean ± SEM. For in vitro analyses, a Shapiro–Wilk test was performed to determine the distribution, and the data were analyzed using ANOVA followed by Dunnett’s test for multiple comparisons between samples and the reference antibiotics. For the in vivo models, statistical analyses were performed using the Kruskal–Wallis-type non-parametric test associated with Dunn’s test for multiple comparisons. All analyses were performed using GraphPad Prism 8 software (Dotmatics, Boston). A *p*-value < 0.05 was considered significant.

## 3. Results

### 3.1. In Vitro Agar Spot Test for Measurement of the Anti-XDR Bacteria Activity of Stool Samples

We first compared the antagonistic effect of the 10 fecal suspensions on the growth of either VRE or OXA-48 CRE relative to pure bacterial cultures and the reference antibiotics used as controls.

The *Escherichia coli* E213 reference strain (negative control) did not inhibit the growth of VRE, in contrast to the *Lactobacillus* spp. S625 reference strain, which exhibited an antagonistic effect similar to that of tigecycline, used as a reference antibiotic. The inhibition zones were 17.15 ± 0.77 mm, 21.93 ± 2.37 mm, and 20.03 ± 0.57 mm for *Escherichia coli* E213, *Lactobacillus* spp. S625, and tigecycline, respectively. All fecal samples had an inhibiting effect against the growth of VRE. The inhibition ratio, normalized to that of tigecycline, was >1 for 6/10 samples, with samples 5 and 6 (S#5 and S#6) showing the largest antagonistic effect. By contrast, samples 2, 4, and 8 (S#2, S#4, and S#8) were less antagonistic than tigecycline, leading to normalized ratios < 1. Despite these variations in profiles, there was no significant difference between any of the fecal samples, the positive biological control, and the reference antibiotic, i.e., tigecycline (Figure 1A).

As for VRE, the *Escherichia coli* E213 reference strain (negative control) did not inhibit the growth of OXA-48 CRE, in contrast to the *Lactobacillus* spp. S625 reference strain, for which the antagonistic effect was similar to that of colistin, used as a reference antibiotic. The inhibition zones were 13.5 ± 2.19 mm, 17.3 ± 0.91 mm, and 18.1 ± 0.75 mm for *Escherichia coli* E213, *Lactobacillus* spp. S625, and colistin, respectively. All fecal samples were able to limit the growth of OXA-48 CRE. The inter-individual variability of inhibition activity was larger than that observed against VRE, with the normalized ratio ranging from 0.85 to 1.7. S#7 and S#8 showed the lowest antagonistic activity against OXA-48 CRE, whereas S#1 and S#5 were significantly more efficient than colistin, i.e., the reference antibiotic (*p* < 0.001 and *p* < 0.0001, respectively), in inhibiting the growth of OXA-48 CRE (Figure 1B). Of note, different fecal samples showed the greatest antagonistic effect against VRE and OXA-48 CRE.

### 3.2. Efficacy of FMT Performed with Selected Fecal Samples on the Intestinal Carriage of XDR in Mice

We then evaluated the ability of FMT to decolonize mice carrying either VRE or OXA-48 CRE in their intestinal tract. The stool samples used for FMT were selected according to their high or low antagonistic effect toward the growth of each bacterium using the agar spot test. S#6 and S#8 (highly and weakly antagonistic, respectively) were used for FMT in the model of VRE carriage (n = 8 per stool sample), while S#5 and S#7 (highly and weakly antagonistic, respectively) were used in the model of OXA-48 CRE carriage (n = 8 per stool sample) (Figure 2A).

FMT was safe, as no alterations in clinical status were observed during the survey. Weight gain was constant for all animals, without any significant difference between animals receiving the three control gavages performed with PBS or the three FMTs, regardless of whether the fecal suspensions displayed high or low growth-inhibiting activity against the bacteria.

On D1 post-gavage, intestinal carriage with either VRE or OXA-48 CRE was confirmed, with a level of detection higher than 10 log_10_ CFU/g of feces for both bacteria in all mice. We observed a continuous decrease in the VRE load from D4 until the end of the experiment, with levels dropping to 3 to 4 log_10_ CFU/g of colon content at D14 or D15. There was no significant difference between mice from the control group and those receiving the FMT, regardless of the stool sample used (Figure 2B,C). For mice colonized with OXA-48 CRE, we observed biphasic kinetics for the controls, with a continuous decrease in bacterial load from D4 to D9, leading to levels of <5 log_10_ CFU/g of feces, followed by a subsequent increase, reaching 7 log_10_ CFU/g in the colon content at D14/D15. FMT performed with S#7, the less antagonistic fecal sample, did not modify the kinetics of OXA-48 CRE carriage in the intestine (Figure 2C). On the contrary, FMT performed with S#5, selected as the most antagonistic fecal sample measured using the agar spot test, led to a significant difference in colonization with OXA-48 CRE at D14/D15 (*p* < 0.0005) (Figure 2B).

## 4. Discussion

Given its effectiveness in the eradication of *C. difficile*, FMT could potentially be used for other indications. It has already been proposed as an alternative strategy to eradicate multidrug-resistant bacteria from the intestinal reservoir. However, mixed results have been obtained in clinical practice, suggesting that the heterogeneity of donor stools may play a role in the patient response. Here, all fecal samples were antagonistic toward the growth of multidrug-resistant bacteria, such as VRE or OXA-48 CRE, in vitro using our agar spot test, but with a varying effect. This result was associated with variability in the efficacy of FMT to reduce or limit intestinal carriage of these bacteria, as only one stool sample selected for its high and significant antagonistic effect, as measured using the agar spot test, was able to significantly lower the intestinal carriage of OXA-48 CRE in mice by approximately 3 log_10_ orders of magnitude within 2 weeks. The agar spot test could thus serve as a simple and reliable screening tool to select stool samples with the best potential to eradicate/limit XDR bacteria carriage after FMT, especially for patients with CRE.

The carriage of multidrug and extensive drug-resistant bacteria is associated with an increased risk of infections by these bacteria for the carriers and a high risk of dissemination in both the healthcare setting and the community. As the gut can become a reservoir for potential antibiotic-resistant pathogens under the pressure of antibiotic treatment, the restoration of a normal microbial community structure using FMT could be a promising approach to decolonize subjects to protect against infections with such pathogens, as well as against *C. difficile*. We previously showed that fecal samples have an antagonistic effect toward the growth of *C. difficile* in vitro and confirmed these results in vivo, showing the efficacy of FMT in restoring a healthy intestinal microbiota with an efficient barrier effect, resulting in the decolonization of *C. difficile* in mice [18]. Here, we developed a similar approach to test the antagonistic effect of stool samples against XDR bacteria. Most fecal suspensions were able to inhibit the in vitro growth of VRE or OXA-48 CRE, selected to be representative of XDR bacteria, with an activity superior or equal to that of one of the reference antibiotics. Interestingly, we observed inhibition profiles that varied between samples. For *C. difficile* eradication, these results confirm the key role of diffusible fermentative metabolites (i.e., short-chain fatty acids, bacteriocins, bile acids, etc.) and/or bacteriophages in the antagonistic effect of the fecal suspensions [21,22,23,24]. Such a role for diffusible products and metabolites has been already suggested in clinical practice, as shown in the pilot study of Ott et al., who showed the positive effect of fecal filtrates in eradicating *C. difficile* in five patients [25].

We then selected the fecal suspensions with the highest and lowest antagonistic effects in vitro to evaluate their ability to eradicate or at least limit the intestinal carriage of VRE or OXA-48 CRE in mice. The mouse model of colonization with XDR has been previously successfully used to examine the effect of antibiotic treatment on the establishment and elimination of intestinal colonization with KPC-Kp [26]. In this study, the administration of non-absorbed oral antibiotics was shown to be an effective strategy to suppress colonization with KPC-Kp. In another study using a mouse model of VRE colonization, treatment with *Lactobacilli* was shown to be a promising approach for reducing VRE colonization in the gut [27]. Although the transferability of the results from the murine model to evaluate the efficacy of FMT in eradicating XDR carriage in patients has never been evaluated, we previously showed the efficacy of FMT in eradicating intestinal carriage of *C. difficile* in mice, with results that corroborated those of clinical studies [18]. As previously shown, treatment of mice for one week with amoxicillin, a broad-spectrum antibiotic, renders mice highly susceptible to VRE and CRE colonization, with approximately 10 log_10_ CFU per gram of feces 24 h after gavage [19,20]. In control mice, the level of colonization decreased to a sub-dominant microbiota level upon termination of the antibiotic treatment, as these bacteria are minor contributors to a colonic microbiota composed predominantly of oxygen-intolerant obligate anaerobes [28,29]. This decrease potentially results from nutrient depletion or quorum-sensing mechanisms that are yet to be determined [19,30]. For OXA-48 CRE, a subsequent increase in colonic load was observed at D14/15. This could be related to the specific biology of *K. pneumoniae* and its ability to invade the mucus layer adjacent to colonic epithelial cells [20]. Under this experimental condition, FMT performed with fecal samples with either high or low antagonistic activity did not modify the profile of intestinal colonization with VRE. This murine model of XDR carriage is a robust model in which the physiology of the animal plays a role, as the caecum can act as a reservoir for XDR that could contribute to the intestinal carriage. The poor efficiency of FMT performed with the human microbiota in the mouse model of intestinal colonization also needs to be analyzed with respect to the absence of a significant effect of various fecal samples based on the agar spot test results relative to reference antibiotics. A randomized clinical study previously showed that an oral decolonization regimen based on reference antibiotics was unable to definitively eradicate XDR carriage in the intestine [31]. It is therefore not surprising that FMT performed with S#7, which was not significantly more effective in vitro than the reference antibiotic, was also ineffective in limiting the intestinal carriage of OXA-48 CRE in vivo. By contrast, the kinetics of colonization with OXA-48 CRE was significantly altered after FMT performed with S#5, which was selected for its significantly higher antagonistic activity in vitro relative to colistin, suggesting that OXA-CRE decolonization can be achieved by complementing the gut microbiota with competing bacteria and/or specific phages. These results confirm a donor effect. It also confirms that, despite similar intestinal localization, the mechanisms underlying resistance to microbiota-mediated colonization are likely distinct for these two different antibiotic-resistant bacterial species.

This was a pilot study based on a limited number of samples. Only samples with extreme effects in vitro were evaluated in vivo to comply with the commonly accepted 3R rule for animal experimentation. Further studies are now required to confirm these results, to identify human microbiota signatures and molecular mechanisms associated with gut decolonization underlying the inhibitory effects of fecal suspensions, and to evaluate the performance of the agar spot test in clinical practice. For clinicians, the agar spot test could represent a simple and reliable screening tool to select stool samples with the best potential for FMT to eradicate/reduce carbapenemase-producing *Enterobacteriaceae* carriage.

## Figures and Tables

**Figure 1 microorganisms-11-02753-f001:**
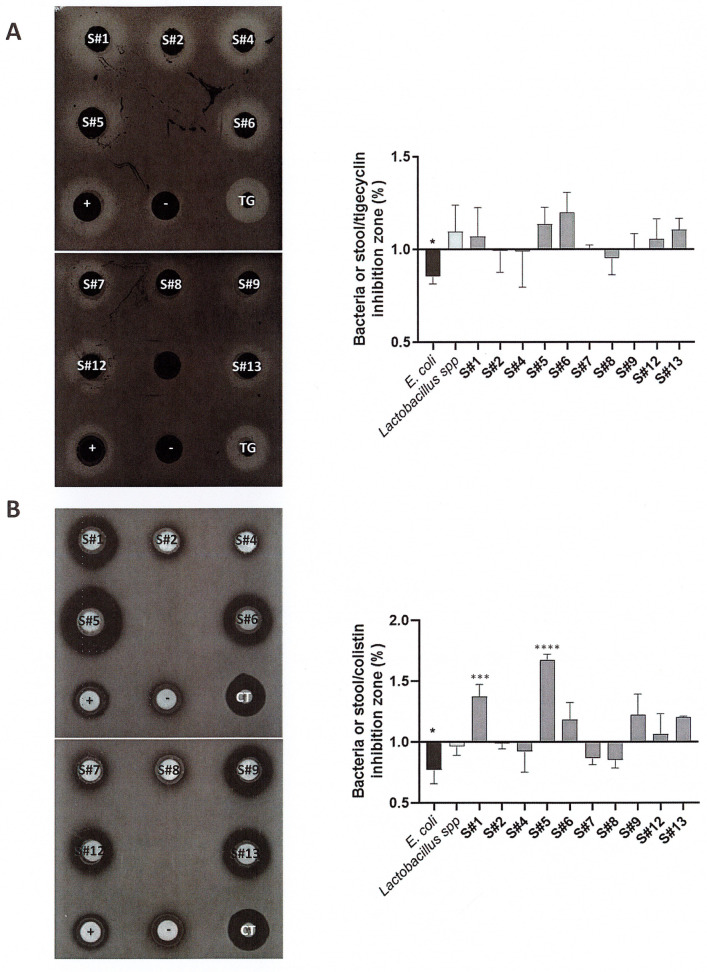
Anti-*E*. *faecium* VanA (**A**) and *K. pneumonia* OXA-48 (**B**) activity of stool samples. Left: representative image of an anti-XDR bacteria agar spot test. Right: results are presented as the ratio between inhibition diameters of pure bacteria or stool samples and the reference antibiotics (e.g., tigecycline (TG) and colistin (CT) for *E. faecium* VanA and *K. pneumonia* OXA-48 agar spot test, respectively) and expressed as mean ± SEM. Statistical significance: * *p* < 0.05; *** *p* < 0.001; **** *p* < 0.0001.

**Figure 2 microorganisms-11-02753-f002:**
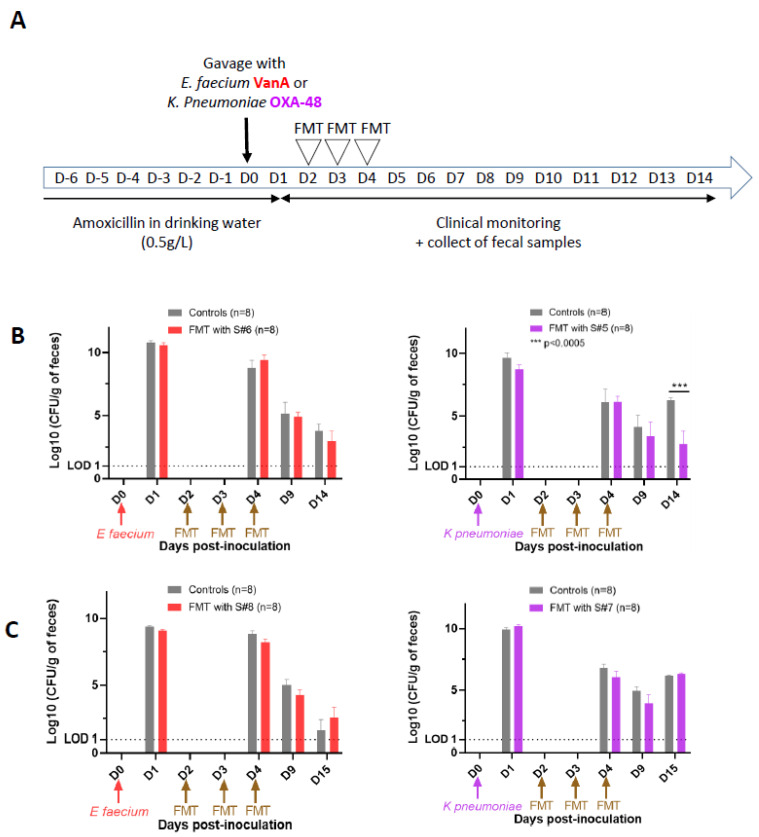
Effect of FMT on intestinal carriage of XDR bacteria. (**A**) Experimental design of the XDR bacteria-colonized FMT mouse model. (**B**) Monitoring of *E. faecium* VanA and *K. pneumonia* OXA-48 fecal load after FMT performed with fecal samples (S#6 and S#5, respectively) highly antagonistic toward in vitro growth of bacteria. (**C**) Monitoring of *E. faecium* VanA and *K. pneumonia* OXA-48 fecal load after FMT performed with fecal samples (S#8 and S#7, respectively) poorly antagonistic toward in vitro growth of bacteria. Results are expressed as mean ± SEM. Statistical significance: *** *p* < 0.0005.

## Data Availability

Data are contained within the article.

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
