# Peer review of "A Simple In Vitro Test to Select Stools for Fecal Microbiota Transplantation to Limit Intestinal Carriage of Extensively Drug-Resistant Bacteria"

_microorganisms, 2023, doi:10.3390/microorganisms11112753_

Round 1

Reviewer 1 Report

Comments and Suggestions for Authors

The manuscript is very interesting and discusses Fecal microbiota transplantation (FMT) as a therapy that restores diverse microbiota and may help to decolonize the intestine from multidrug-resistant bacteria such as VRE and OXA-48 CRE.

Some concerns are raised in this manuscript.

1- Could you please clarify why you exclusively relied on the Agar spot test as a diagnostic tool to assess antagonistic activity against VRE and OXA-48 CRE in vitro? Were there particular reasons for not incorporating molecular methods into your analysis, and how do you account for the observed inter-individual variations?

2- please provide references or additional information to support the choice of using a mouse model for intestinal colonization as a preclinical approach to evaluate the efficacy of FMT with specific fecal samples in reducing or eliminating VRE and OXA-48 CRE colonization in vivo.

3- The manuscript needs linguistic editing in your manuscript to improve its clarity and readability

Comments on the Quality of English Language

The manuscript needs linguistic editing in your manuscript to improve its clarity and readability

Reviewer 2 Report

Comments and Suggestions for Authors

The article discusses the issue of using fecal microbiota transplantation to decolonize bacterial strains with extended-spectrum antibiotic resistance (XDR) from the human gastrointestinal tract. The article discusses several main issues related to:
• Determination of the ability of human microbiota to inhibit the growth of multidrug-resistant microorganisms (on the example of K. pneumoniae producing OXA-48 carbapenemase and E. faecium resistant to vancomycin VRE)
• Determining whether this ability depends on the donor
• Determining whether the microbiota is equally useful for decolonizing bacteria with different types of antibiotic resistance
• Proposing a test and determining whether this test can be useful in predicting the ability of a given microbiota to decolonize ARB strain.

To try to answer these questions, the authors describe a simple in vitro test that allow to answer the question of whether the human microbiota can inhibit the growth of pathogens possessing the above-mentioned resistance mechanisms and, if so, whether there are differences in these abilities between microbiota from different donors. Then, using the results obtained in the in vitro test, they compare them with an in vivo decolonization attempt on a mouse model. During the in vitro test, the authors state that the human microbiota has the ability to inhibit the growth of both E. faecium VRE and K. pneumoniae OXA-48, but, in the case of VRE, no statistically significant differences were found between the tested stool samples, controls and the reference antibiotic, and in the case of OXA-48, these differences were significant in two stools compared to the reference antibiotic. At the same time, they note that the feces that were characterized by the greatest antagonism towards E. faecium VRE were different from those that inhibited the growth of K. pneumoniae OXA-48.
Based on the results obtained in this test, two stools were selected for each of the tested resistance mechanisms, each of which showed a high and low antagonistic effect towards the reference antibiotic. Then, a group of mice was first administered an antibiotic, and then, at the end of antibiotic treatment, a suspension containing E. faecium VRE or K. pneumoniae OXA-48 strains was administered. The day on which the suspension was administered to the animals was marked as day D0, and the following days were marked with the letters D and a number indicating the number of days since the administration of the bacterial strains. On days D2, D3 and D4, a transplant of intestinal microbiota obtained from stools selected for a given resistance mechanism (with high and low antagonistic effect) or from PBS buffer was carried out as part of the control group.
Then, mouse feces from days D1 (before FMT), D4, D9 and D14 or D15 were collected, and the content of K. pneumoniae OXA-48 or E. faecium VRE strains were determined, respectively.
In the case of E. faecium VRE, no significant changes were found in the content of strains compared to mice that received the PBS buffer, while in the case of K.pneumoniae OXA-48, a statistically significant decrease in the content of strains was found, but only for feces, which in the in vitro test showed a high antagonism.
Based on the results of these studies, it was concluded that feacal microbiota transplantation may be effective in drug-resistant strains decolonization, and the in vitro test itself may be a useful tool in selecting specific stools (microbiota) for the specific ARB decolonization.
The authors made interesting observations indicating large differences between different microbiota compositions in the ability to decolonize the GI tract that these abilities vary depending on the given resistance mechanism and/or strain. Therefore, they opened a new door for further research and looking for answers to what makes a given microbiota composition capable of decolonizing a given strain with resistance mechanism. The simple test presented by the authors may constitute an easy tool for predicting the ability to decolonize the ARB until more precise tools are developed to answer these questions.
Bigger issues:
• the authors do not clearly present the method of obtaining stool donors and the stool itself (page 3, lines 2-4) - what was the criteria of stool/microbiota selection?
• the selection of donors for the study is very poor and raises concerns that the microbiota used may have been of low quality (page 3, lines 4-8). I see that this was the purpose to distinguish a decolonization potential between stools, but more technical information should be provided,
• the authors do not check the effectiveness (or do not provide information about it) of the viability of thawed microbiota, either qualitatively or quantitatively, which may have a key impact on its effectiveness in decolonization abilities (page 3, line 22-)
• based on which criteria it was considered that the microbiota has an antagonistic effect on the growth of microorganisms with resistance mechanisms (how much larger the zone of inhibition must be to consider this statement true) (page 5, lines 4-14)?

Minor issues:
• use of the genus name "Clostridium" instead of "Clostridioides" (page 2 line 23)
• no source given for the statement that the two the two most common extended-resistant (XDR) bacteria causing infections in hospitalized patients are K. pneumoanie OXA-48 type and E. feacium VRE (page 2, lines 46-51)
• lack of clear presentation of the method of preparation of collected feces (page 3, lines 4-8)

Comments on the Quality of English Language

English should be slightly improved

Reviewer 3 Report

Comments and Suggestions for Authors

Dr. Salandre and her colleagues have developed an agar spot test to evaluate the screening performance of a simple diagnostic tool for selecting fecal samples effective at decolonizing the intestine. Fecal samples from 10 healthy subjects were selected in a pilot study. This spot test was developed to evaluate their antagonistic activity toward the growth of VanA Enterococcus faecium and OXA48-producing Klebsiella pneumoniae.

Their approach shows promise, nevertheless, several issues need to be addressed:

1. Introduction

Minor 1.1: The authors should also mention the efficiency of Fecal Microbiota Transplantation (FMT) in primary severe Clostridioides difficile infections. In fact, the overall cure rate is over 85%. You could use Dr. Popa and colleagues’ work or any other relevant paper as a reference and citation:

[ref] Popa, D.; Neamtu, B.; Mihalache, M.; Boicean, A.; Banciu, A.; Banciu, D.D.; Moga, D.F.C.; Birlutiu, V. Fecal Microbiota Transplant in Severe and Non-Severe Clostridioides difficile Infection. Is There a Role of FMT in Primary Severe CDI? J. Clin. Med. 2021, 10, 5822. https://doi.org/10.3390/jcm10245822

Minor 1.2: The authors need to update the terminology to "Clostridioides difficile" instead of "Clostridium difficile."

Major 1.3: They need to mention from the very beginning that they are proposing a pilot test (abstract, introduction, and first and foremost in the materials and methods section). The reader finds out about the nature of this study only in the discussion section..

2. Materials and Methods

Major 2.1: How did the authors approach the analysis of the distribution, given such a small sample, before performing the ANOVA test and Kruskall-Wallis? This should be properly addressed in the Methods section.

Major 2.2: What was their chosen statistical significance level? Was it 0.05 or 0.1?

3. Discussion

Major: The authors should properly address the limitations of this study. The most significant one is the sample size. Additionally, even though they mention a previous work showing that fecal samples had an antagonistic effect toward the growth of C. difficile in vitro, it is not clear how this current approach with an agar spot test is similar to their previous report [ref 14]. Moreover, the fact that this agar spot test didn’t include a C. difficile strain along with the XDR bacteria should be presented as a limitation of their study, given that an important part of their discussion section is focused on conveying the idea of FMT effectiveness against C. difficile, especially since this test might be useful for clinicians performing FMT.

Minor: The last paragraph should be rephrased for clarity. Perhaps, "For clinicians, the agar spot test could represent a simple and reliable screening tool to select stool samples with the best potential for FMT to eradicate/reduce carbapenemase-producing Enterobacteriaceae carriage."

Comments on the Quality of English Language

Moderate editing of English language required

Round 2

Reviewer 3 Report

Comments and Suggestions for Authors

The authors have properly addressed all the raised issues.

The manuscript has been significantly improved and is ready for publication.